# Effect of Grain Size on the Plastic Deformation Behaviors of a Fe-18Mn-1.3Al-0.6C Austenitic Steel

**DOI:** 10.3390/ma15248717

**Published:** 2022-12-07

**Authors:** Ziyi Cui, Shudong He, Jie Tang, Dingfa Fu, Jie Teng, Fulin Jiang

**Affiliations:** College of Materials Science and Engineering, Hunan University, Changsha 410082, China

**Keywords:** TWIP steel, grain size, work hardening, dislocation, twinning

## Abstract

Grain size is a microscopic parameter that has a significant impact on the macroscopic deformation behavior and mechanical properties of twinning induced plasticity (TWIP) steels. In this study, Fe-18Mn-1.3Al-0.6C steel specimens with different grain sizes were first obtained by combining cold rolling and annealing processes. Then the influence of grain size on the plastic deformation mechanisms was investigated by mechanical testing, X-ray diffraction-based line profile analysis, and electron backscatter diffraction. The experimental results showed that the larger grain size could effectively promote twinning during plastic straining, produce an obvious TWIP effect, and suppress the rate of dislocation proliferation. The continuous contribution of dislocation strengthening and twinning functions led to a long plateau in the work-hardening rate curve, and increased the work-hardening index and work-hardening ability. At the same time, the strain could be uniformly distributed at the grain boundaries and twin boundaries inside the grain, which effectively relieved the stress concentration at the grain boundaries and improved the plasticity of deformed samples.

## 1. Introduction

As some of the most promising automotive steels, twinning induced plasticity (TWIP) steels possess excellent properties of high strength, high plasticity and high energy absorption capacity [1,2], which are closely related to its twinning behavior and dislocation slip characteristics during work-hardening [3,4]. However, over the years, there have been many arguments regarding the work-hardening mechanism of TWIP steels, such as the twin hardening mechanism [5,6,7], dislocation hardening mechanism [8], atomic cluster hardening mechanism [9], and integrated hardening mechanism [10]. Due to the simultaneous action of twinning and high dislocation density during straining, it is difficult to accurately assess their individual strengthening contributions and their effect on the mechanical properties of steel during the various stages of work-hardening. Among the factors affecting the twinning behavior and dislocation proliferation in high-manganese TWIP steels, grain size plays an important role [11].

Fine-grain strengthening, which is a strengthening method that can simultaneously improve the strength and toughness of metallic materials, has a great impact on the overall performance of high-manganese austenitic steels. It is widely accepted that, to a certain extent, the mechanical properties improve with a decrease in grain size and an increase in grain boundary area, as grain refinement has a better strengthening effect on metallic materials [12,13]. High-manganese steel can be annealed after cold deformation to obtain different grain sizes by controlling the annealing temperature and holding time. There are still relatively few studies on the effect of grain size on the TWIP effect of high-manganese steels. In the work of Sang et al. [14], starting from the effect of grain size on the stacking fault energy, it was found that the stacking fault energy and the critical nucleation stress of twinning decreased with increasing grain size, resulting in a lower twinning difficulty and a higher density of twinning in coarse-grained TWIP steels at the same strain level. Ueji et al. [15] also found the inhibition of twinning by fine grains in tensile experiments on three different grain sizes of high-manganese steel samples. However, some researchers [16] have argued that fine grains promoted more twinning because of the uniform dislocation distribution in fine grains, which provided favorable conditions for the dislocation reaction during twinning nucleation. In contrast, dislocations in coarse grains showed a cell structure that was not uniformly distributed, providing fewer sites for twin nucleation. Therefore, they were more likely to form low-density secondary twinning systems. In general, the direct effect of grain size on the TWIP effect has not been authoritatively established and remains to be studied in-depth.

In high-manganese steels, the twinning behavior during plastic deformation affects its work-hardening properties, and the grain size affects the work-hardening properties by influencing the difficulty of twinning. Some researchers [14,15,16] illustrated the relationship between grain size and work-hardening behavior through the effect of grain refinement on twinning inhibition and back stress, and the back stress relationship was shown in Equation (1) [4,15,16]:(1)$$\sigma_{B}=MGbn/D$$
where *n* is the number of dislocations that are stacked due to hindered dislocation slip and *D* is the effective grain size.

As can be seen in Equation (1), a decrease in *D* leads to an increase in back stress, which makes the width of the stack fault narrower, inhibits the planar slip of dislocations, and promotes the cross slip of dislocations, making twinning more difficult. The results of Ueji et al. [15] also confirmed that twins were more likely to form in large grains, and the high density of twins enhanced the pinning effect on dislocations, resulting in higher work-hardening properties of coarse-grained high-manganese steels. In most of the previous studies [12,13,14,15,16,17,18], the focus has been on the effect of grain size on the work-hardening behavior, and less attention has been paid to the effect of the interaction of twinning behavior and dislocation proliferation during various stages of work-hardening.

Therefore, in this paper, the effects of twinning behavior and dislocation proliferation on the work-hardening behavior of TWIP steel with different grain sizes were studied. Tensile testing was used to investigate the trend in mechanical properties and work-hardening behavior of rolled samples with different grain sizes. The phase analysis was performed using XRD analysis, and the dislocation density calculation was carried out by XRD line profile analysis. The twinning behavior and plastic strain distribution were analyzed using electron backscatter diffraction (EBSD) to reveal the influence of grain size on the work-hardening mechanism of high-Mn TWIP steel.

## 2. Experimental Procedures

The high-manganese steel composition chosen for this experiment was Fe-18Mn-1.3Al-0.6C (wt.%). The hot rolled steel was solid solution treated at 1100 °C for 1 h, followed by water quenching to obtain single-phase austenite organization. The steel is then cut into 10 mm thick sheets using a wire-electrode cutter and cold rolled at room temperature with a total reduction of 40%. Then different annealing treatments were adopted to obtain four different recrystallized grain sizes; the annealing temperatures and times were 700 °C for 0.5 h, 800 °C for 1 h, 850 °C for 1 h, and 900 °C for 4 h. Four groups of samples with average grain sizes in the range of 5~10 μm, 10~30 μm, 50~70 μm, and 100~200 μm were obtained, as shown in Figure 1. Based on the quantified statistics from Image Pro Plus software, the average grain size was 6.8 μm after 0.5 h annealing at 700 °C, 27.2 μm after 1 h annealing at 800 °C, 60.3 μm after 1 h annealing at 850 °C, and 103.4 μm after 4 h annealing at 900 °C. In addition, few annealing twins were observed, which had little effect on the mechanical properties of the annealed samples due to the very small volume fraction. Then the samples with different grain sizes were further rolled at room temperature with a 10%, 20%, 30%, and 40% reduction, respectively. The experimental processes are shown in Figure 2.

Tensile specimens (size:111 mm^L^ × 15 mm^W^ × 3 mm^T^) were machined from cold rolled sheets by wire-electrode cutting. Tensile tests were then carried out at a strain rate of 6 × 10^−3^ s^−1^ using an Instron-3382 universal tensile testing machine (Instron Corporation, Norwood, MA, USA), where the tensile specimens had the same tensile direction as the rolling direction. Phase and dislocation density analysis of cold rolled sheets were characterized by XRD technique (SmartLab SE). The samples used in the XRD experiment were mechanical and electrolytically polished to fully remove the mechanically affected layers [19,20,21,22,23,24]. XRD measurements were performed by using Cu-Kα radiation at a scanning speed of 1.2°/min with a 2θ range from 40° to 140°. For tensile test and XRD, three samples were checked for each group and the average results and standard error bars were presented. The operating voltage and current were 40 kV and 40 mA, respectively. Microstructural analysis of experimental steels was carried out by means of an optical microscope (OM, Shanghai Metallographic Machinery Equipment Co., Ltd., Shanghai, China) and scanning electron microscope (SEM, JSM-7900F, JEOL Ltd., Tokyo, Japan) which was equipped with an electron backscatter diffraction camera (EBSD, Oxford Instruments, Abingdon, UK). OM samples were mechanically polished and then etched with a 7% nitric acid and 93% ethanol solution. EBSD samples were prepared by mechanical polishing, followed by electrolytic polishing. The electrolytic polishing was carried out in a mixture of 10% perchloric acid and 90% acetic acid using a voltage range of 30–40 V for 40 s.

## 3. Results and Discussion

### 3.1. Tensile Properties

The engineering stress-strain curves of four groups of rolled samples with different grain sizes are shown in Figure 3. The curves are smooth and without waviness, and no dynamic strain aging effect occured. The tensile strengths of the four groups of rolled samples with different grain sizes increased with the amount of reduction, while the plasticity decreased. The tensile properties of the samples with different grain sizes showed different trends. The sample with an average grain size of 6.8 μm exhibited good plasticity at a lower reduction, with a tensile strength of 1324 MPa and elongation of 60%, which was the best among the four groups of samples. However, the plasticity decreased sharply with the increase in the rolling reduction, and only 13.1% elongation was achieved at a 40% reduction, which indicates a high work-hardening rate and a poor work-hardening capacity. It is worth noting that the plasticity of the sample with grain size 103.4 μm was lower than that of other three groups of samples at a low reduction, and the elongation was approximately 50%. With the increase in reduction, its plasticity decreased at a significantly slower rate compared with the other three groups of samples. The elongation was still 22% when the reduction increased to 40%.

### 3.2. Effect of Grain Size on Work-Hardening Behavior

The grain sizes could influence twinning and dislocation proliferation, resulting in different work-hardening behaviors. In this section, the work-hardening behaviors of the four groups of rolled samples with different grain sizes was analyzed to investigate the way in which the differences in twinning and dislocation proliferation rate caused by different grain sizes affected the TWIP effect at various stages of plastic deformation. The effect on the work-hardening capacity was also investigated. The true stress σ and true strain ε were calculated from the engineering stress $\sigma^{\prime }$ and engineering strain $\varepsilon^{\prime }$ by the following equations:(2)$$\sigma =\sigma^{\prime }\left(1+\varepsilon^{\prime }\right)$$
(3)$$\varepsilon =ln\left(1+\varepsilon^{\prime }\right)$$

Then, the true stress-strain curve was plotted and the derivative of the true stress with respect to true strain was used to obtain the work-hardening rate curve, as shown in Figure 4, where the corresponding true stress-true strain curve is shown in the small figure.

The work hardening rate of the annealed samples with four different grain sizes generally increased with increasing rolling reduction and decreased with increasing grain size, which was due to the work hardening caused by the plastic strain exerted by rolling and the effect in the subsequent tensile experiments. As is shown in Figure 4, in the four samples with different grain sizes and 10% rolling reduction, the degree of work hardening caused by rolling was not high due to the small degree of pre-deformation. Furthermore, four stages of work hardening could be clearly observed in the work hardening rate curves:

Stage Ⅰ: At a low true strain (0.04 < ε < 0.09), the work-hardening rate decreases sharply to about 2500 MPa (103.4 μm sample) and 3000 Mpa (6.8 μm sample), in which the deformation mode of the sample changes from elastic to plastic deformation, and the recovery phenomenon occurs in the low-strain level. The dislocation density decreases slightly during recovery stage, which causes the work-hardening rate to change from a sharp decrease to a slow decrease [25].

Stage Ⅱ: In this stage (0.09 < ε < 0.15), the increase in plastic strain leads to a large multiplication of dislocations and starts to slip. In addition, the sample undergoes macroscopic deformation, at which time the work-hardening rate is reduced slowly or no longer reduced by dislocation strengthening, and a plateau appears [26]. Then the work-hardening rate reaches a minimum value of 2400 Mpa (103.4 μm sample) to 2800 Mpa (6.8 μm sample).

Stage Ⅲ: With the gradual increase in true strain (0.15 < ε < 0.35), the stress on the sample reaches the critical nucleation stress of twinning. The twin starts to generate and proliferate, which acts as a barrier to the dislocation slip, and the dislocation forms a large amount of dislocation tangle at the twin boundary [27,28,29]. The strain hardening rate gradually picks up and reaches the maximum value of 3000 Mpa (103.4 μm sample) to 3500 Mpa (6.8 μm sample).

Stage Ⅳ: With the further increase in true strain (ε > 0.35), the plastic strain continues to increase. The volume fraction of twins increases to the saturation value, which cannot hinder the continuous proliferation of dislocations and slip. Then the crack gradually expands, and the strain hardening rate continues to decrease until the plastic deformation process ends and the sample fractures [4].

With the increase in the rolling reduction, the dislocation density and the volume fraction of twin inside the samples differed due to the different grain sizes. Therefore, the stage of work hardening of each sample showed different characteristics, and the three groups of samples with 20%, 30%, and 40% of the rolling reduction have been discussed separately.

As is shown in Figure 5, two different strain hardening modes were observed in the four groups of samples with a 20% reduction, in which the large grain size samples (103.4 μm, 60.3 μm) maintained the complete four-stage hardening pattern. In contrast, the two smaller grain size samples (6.8 μm, 27.2 μm) only showed stages Ⅰ, Ⅱ, and Ⅳ (without stage Ⅲ), which were affected by the TWIP effect due to twinning. This is because the twin volume fractions of the samples with different grain sizes differed under larger pre-deformation, and the samples with smaller grain sizes were already saturated with twin during pre-deformation due to their lower saturation volume fraction of twins.

With the further increase in rolling reduction, as is shown in Figure 6, it can be seen that, in the four groups of samples with a 30% reduction, only work-hardening stages Ⅰ and Ⅳ were observed in the samples with a 6.8 μm grain size. However, the three stages of Ⅰ, Ⅱ, and Ⅳ were still observed in the samples of the three groups with a larger grain size, due to the larger grain boundary area in the small size grains accelerating dislocation proliferation and enhancing blocking functions. In addition, the dislocation strengthening was already over in the pre-deformation stage of rolling, so the work-hardening rate decreased rapidly until the sample fractured.

As can be seen in Figure 7, only the 60.3 μm and 103.4 μm grain size samples retained stage Ⅱ. However, only stages I and Ⅳ of strain hardening were observed in the remaining two groups of samples, indicating that, as the grain size increases, twinning is promoted during the pre-deformation process. In addition, the dislocation proliferation and tangling stages lasted longer; thus, stage Ⅱ was strengthened and prolonged, and the work hardening ability was enhanced.

The work-hardening index is a parameter commonly used to indicate the work-hardening ability of materials, also known as the instantaneous deformation hardening index, which can be calculated by the Hollomon formula to further investigate the work-hardening ability of rolled parts with different grain sizes under different pressures [30]. The Hollomon formula is as follows:(4)$$\sigma =K\varepsilon^{n}$$
where K is the strength factor and *n* is the work-hardening index; the value of *n* can be obtained by taking the logarithm of both sides of the equation and deriving it as shown in Equation (5).
(5)$$n=\frac{\varepsilon }{\sigma }\frac{d\sigma }{d\varepsilon }$$

According to the work-hardening index *n* calculated in Equation (5) of the four groups of samples with different grain sizes, the variation in *n* values for the samples with different rolling reductions is plotted, as shown in Figure 8. It can be seen that the work-hardening index of the four groups of samples with different grain sizes gradually decreased as the reduction increased. The TWIP effect was lower in the large grain size samples due to the lower dislocation density and less twinning in the sample at a low reduction. The *n* value of the sample with the average grain size of 6.8 μm was 0.8, while the *n* value of the sample with the average grain size of 103.4 μm was 0.65. With the increase in the rolling reduction, the internal dislocations of the fine-grain samples proliferated rapidly, which resulted in a rapid decrease in the work-hardening index. The volume fraction of twins in the large grain size sample was higher than that of the small grain size sample at a higher pre-deformation, which promoted the TWIP effect effectively and resulted in the largest work-hardening index of 0.25 for the 103.4 μm grain size sample when the rolling reduction was increased to 40%. However, the work-hardening index of the 6.8 μm grain size sample decreased rapidly to 0.12, which was the lowest among the four groups of samples.

### 3.3. XRD Analysis and Dislocation Density Calculation

To investigate the effect of dislocation multiplication on the work-hardening behavior of the samples with different grain sizes, XRD analysis was first performed. The XRD patterns of rolled samples with different grain sizes are shown in Figure 9. According to the diffraction pattern, the samples still had single-phase austenite microstructure after plastic deformation, and no martensite phase transformation occurred. Therefore, the strengthening mechanism in the experimental steel was mainly the TWIP effect, but not the TRIP effect. As can be seen in b, the intensity of some diffraction peaks decreased significantly with the increase in plastic strain, and the diffraction peaks were shifted and broadened during the plastic deformation process. This corresponds to the lattice strain behavior during the elastic-plastic deformation of the experimental steel, where the elastic lattice strain causes a shift of the diffraction peaks and the plastic lattice strain causes a line profile broadening of the diffraction peaks [19,22,31].

To fit these behaviors, Warren et al. [31] used a Fourier transform ($A\left(L\right)$) to calculate the line profiles:(6)$$lnA\left(L\right)=lnA^{s}\left(L\right)-lnA^{D}\left(L\right)=lnA^{s}\left(L\right)-2\pi^{2}L^{2}\left[\varepsilon {\left(L\right)}^{2}\right]/d^{2}$$
where$\;A^{s}\left(L\right)$ is the Fourier coefficient of the grain size, $A^{D}\left(L\right)$ is the Fourier coefficients of lattice distortion, L is the Fourier parameter, and [$\varepsilon {\left(L\right)}^{2}$] describes the lattice strain with a function of:(7)$$\left[\varepsilon {\left(L\right)}^{2}\right]=\left(\rho \bar{C}b^{2}/4\pi \right)ln\left(R_{e}/L\right)$$
where ρ, b, and$\;\bar{C}$ represent the density, the magnitude of the Burgers vector, and the average contrast factor of dislocations [19,20,21], respectively. $R_{e}$ is the effective outer cut-off radius of dislocations.

As mentioned before, several effective line profile analysis methods have been developed. In this work, the modified Williamson–Hall and modified Warren–Averbach (mWH/mWA) methods were used to estimated the dislocation densities of the plastic deformation samples. The contributions of planar defects could be separated by improving the mWH method:(8)$$\Delta K=\frac{0.9}{D}+\frac{\pi B^{2}b^{2}}{2}\rho^{\frac{1}{2}}\left(KC^{\frac{1}{2}}\right)+O\left(K^{2}C\right)$$
where ΔK=βcosθ/λ, K=2sinθ/λ, θ and λ are the diffraction angle and wavelength of X-rays, β is the full width at half maximum (FHWM) or integral breadth, and *D* is the grain size. *C* is the dislocation contrast factor, which is utilized for correcting strain anisotropy and introducing *q* as a constant related to elastic constants and character of dislocations. *C* is related to *q* by the following equations:(9)$$\bar{C_{hkl}}=\bar{C_{h00}}+\left(1-qH^{2}\right)$$
(10)$$H^{2}=\frac{h^{2}k^{2}+k^{2}l^{2}+l^{2}h^{2}}{h^{2}+k^{2}+l^{2}}$$
where $H^{2}$ is the orientation parameter and *h*, *k*, *l* are the Miller indices. A program based on the least squares method was designed to estimate the experimental values of *C_hkl_*, and *q* by the best fitting results of the mWH plots. The fitting results are shown in Figure 10a,b.

Then dislocation density can be quantitatively estimated by the mWA method based on the Fourier analysis, as given below:(11)$$lnA\left(L\right)=lnA^{s}\left(L\right)-\frac{\pi b^{2}}{2}\rho L^{2}\ln\left(\frac{R_{e}}{L}\right)\left(K^{2}C\right)+O{\left(K^{2}C\right)}^{2}$$

Thus, the slope $Y\left(L\right)$ can be obtained, and its expression is shown as follows:(12)$$\frac{Y\left(L\right)}{L^{2}}=-\left(\frac{\pi b^{2}}{2}\rho \right)lnR_{e}+\left(\frac{\pi b^{2}}{2}\rho \right)lnL$$

In summary, a plot of $Y\left(L\right)/L^{2}$ and lnL in the mWA-based method for experimental steels with different plastic deformation methods under different strain is obtained, as shown in Figure 10c, and the slope of its trend line is the estimated dislocation density. The results of the dislocation density estimated by the mWH/mWA method are shown in Figure 10d. It can be seen that the samples with a smaller average grain size have a higher dislocation density due to the increase in grain boundary area and accelerated dislocation proliferation and tangling at a 10% rolling reduction. The dislocations in the samples with the average grain size of 6.8 μm and 27.2 μm proliferated rapidly and reached near saturation with the increase in the reduction to 20%; therefore, the dislocation density growth rate of these two rolled samples decreased significantly when the reduction amount continued to increase to 30%. The disappearance of the work-hardening stage II in the work-hardening behavior is shown, which is consistent with the results of the work-hardening rate curves in Figure 4.

### 3.4. Microstructural Characteristics of Deformed Steels

In order to investigate the effect of the microstructure in annealed samples with different grain sizes on the work-hardening behavior after pre-deformation, electron backscatter diffraction (EBSD) was also conducted on the rolled samples with different grain sizes under 10%, 20%, and 30% reductions, respectively. The results are shown in Figure 11. After annealing and rolling, annealed twins and mechanical twins were produced inside the samples, which had an orientation difference of 60° from <111>, and they were ∑3 co-grid twins, which are marked by red lines on the image quality (IQ) maps [32,33].

From the inverse pole figure (IPF) and IQ maps, it can be seen that the twinning behavior of the annealed samples varied considerably with grain size. At a 10% reduction, only a small number of annealed twins existed in the three groups of smaller grain sizes (6.8 μm, 27.2 μm, and 60.3 μm), and the grain morphology was intact and clear. While the average grain size of 103.4 μm started to deform in the rolling direction, with slip bands and mechanical twinning. However, the volume fraction of the twins in the four groups of samples was not significantly different at a 10% reduction due to the presence of a small number of annealed twins in the other three groups of samples. With the increase in the rolling reduction to 20%, the average grain size samples of 60.3 μm and 103.4 μm showed a large increase in internal twins and the volume fraction of twins increased rapidly. The two groups of samples with an average grain size of 6.8 μm and 27.2 μm showed an increase in slip lines, and there were a small number of mechanical twins with low twin density. When the reduction was increased to 30%, many mechanical twins were seen in the samples with an average grain size of 27.2 μm, 60.3 μm and 103.4 μm, and the grains were significantly deformed along the rolling direction. Activated secondary twinning was observed in the sample with grain size of 103.4 μm. The average grain size of 6.8 μm was not significantly deformed along the rolling direction, and the number of twinned grains was still few and scattered throughout the sample. The fine grain size increased the critical nucleation stress for mechanical twinning and thus hindered the development of mechanical twinning, satisfying a relationship similar to the Hall-Petch relationship [34]:(13)$$\sigma_{T}=\sigma_{T0}+K_{T}d^{-A}$$
where $\sigma_{T}$ is the mechanical twinning critical nucleation stress, $\sigma_{T0}$ is the lattice friction, $K_{T}$ is the coefficient related to the average grain size *d*, and *A* is a constant (0.5 < *A* < 1).

Equation (13) shows that the critical nucleation stress for mechanical twinning is inversely proportional to the grain size; when the grain size was decreased from 103.4 μm to 6.8 μm in this experiment, the value of ($\sigma_{T}$ − $\sigma_{T0}$) increased approximately 4~15 times due to the decrease in the average grain size *d*. This makes it more difficult to nucleate mechanical twins in small-sized grains, and mechanical twins are generated later in the plastic deformation process and are suppressed at a higher plastic strain. In summary, the increase in grain size can effectively promote the generation and proliferation of mechanical twins, thus contributing to the TWIP effect, increasing the plasticity of the sample and enhancing the work-hardening ability.

The plastic strain distribution of the samples was analyzed by Kernel average misorientation (KAM) maps, and the strain of the sample with an average grain size of 6.8 μm was mainly distributed at the grain boundaries at a 10% rolling reduction. The stress concentrations occurred at the aggregation of fine grains, while the two groups of samples with average grain size of 27.2 μm and 60.3 μm had a more uniform distribution of strain inside the grain. Futhermore, the sample with an average grain size of 103.4 μm had a gradual accumulation of plastic strain at the twinning boundary inside the grain due to the generation of twins. The plastic strain of the sample with an average grain size of 6.8 μm still concentrated at the grain boundaries as the rolling reduction increased, and was gradually distributed at the twin boundaries and slip lines inside the grains due to the generation of slip lines and mechanical twins at 30% reduction. As the grain size increased, the twin generation and proliferation sped up, and the plastic strain was distributed more rapidly and uniformly at the twin boundaries inside the grain with the twin generation, which effectively relieved the stress concentration and thus improved the plasticity of the sample.

The distribution of KAM values can be observed visually by the distribution of strains in samples with different grain sizes under different rolling reductions [35]. The number fraction of KAM values at high angles increased as the strain increased and the plastic lattice strain increased, as shown in Figure 12.

As can be seen in a, in the two groups of samples with an average grain size of 27.2 μm and 60.3 μm, the number fraction of KAM values was concentrated in the low-angle orientation difference (0°~1°) because of the uniform strain distribution. However, in the sample with an average grain size of 6.8 μm, the distribution of KAM values widened and gradually moved toward the high-angle orientation difference because the strain was concentrated at the grain boundaries where the fine grains were clustered. Similarly, the strain in the sample with an average grain size of 103.4 μm showed stress concentration due to the generation of fine twins, and the same broadening of the KAM value distribution was observed. With the increase in the reduction to 20%, as is shown in b, many mechanical twins started to appear in the sample with an average grain size of 60.3 μm, causing the strain to gradually move from a uniform distribution to the grain and twin boundaries. Therefore, a significant broadening of the KAM value distribution towards a high orientation difference of >2° could be seen. At a 30% reduction, the strain continued to accumulate at the grain boundaries due to the lower volume fraction of twin integrals in the sample with an average grain size of 6.8 μm. The samples with an average grain size of 60.3 μm and 103.4 μm gradually distributed the plastic strain from the grain boundaries to the intra-grain twin boundaries due to the further proliferation of twins and activation of the secondary twinning system, and they became more uniform. As a result, the tendency of KAM values to be distributed to higher angle orientation differences was slowed down.

## 4. Conclusions

In summary, mechanical testing, XRD analysis, and EBSD observation were performed on the rolled Fe-18Mn-1.3Al-0.6C austenitic steel samples with different grain sizes to investigate the influence of grain size on twinning behavior, dislocation proliferation, and KAM distribution, as well as the role of such an influence on their overall mechanical properties and work-hardening behavior. The following conclusions were obtained:The tensile test results showed that the plasticity of the annealed samples decreased with the increase in grain size with a less than 10% rolling reduction. However, with the increase in rolling reduction, the plasticity of the small-sized grain samples decreased rapidly, and the sample with an average grain size of 6.8 μm had only a 13.1% elongation at a 40% reduction. The plasticity of the sample with an average grain size of 103.4 μm still achieved a 22.3% elongation at a 40% reduction.By combining XRD analysis for dislocation density calculation and electron backscatter diffraction observation, it could be concluded that a larger grain size could slow down the dislocation proliferation. Since the critical nucleation stress of mechanical twinning decreased with increasing grain size, large grain size samples could effectively promote twinning generation and growth in plastic strain, producing an obvious TWIP effect. The contribution of continuous dislocation strengthening and twinning led to a longer plateau period in the work-hardening rate curve and increased the work-hardening index and work-hardening ability.The KAM maps and their distribution analysis showed that in the samples with an average grain size of 6.8 μm, the strain was always concentrated at the grain boundaries, and a small number of strains accumulated at the twin boundaries due to the generation of twins at a high reduction. In the three samples with larger grain sizes, the strain was uniformly distributed at the grain boundaries and twin boundaries inside the grains due to the extensive generation and proliferation of twins, which effectively relieved the stress concentration at the grain boundaries and improved plasticity.

## Figures and Tables

**Figure 1 materials-15-08717-f001:**
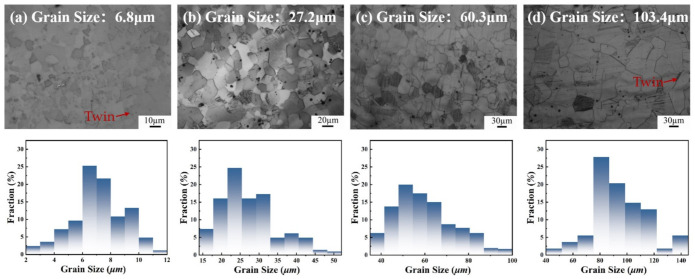
OM and grain size statistics of Fe-18Mn-1.3Al-0.6C steel after cold rolling and annealing treatment: (**a**) 700 °C, 0.5 h; (**b**) 800 °C, 1 h; (**c**) 850 °C, 1 h; (**d**) 900 °C, 4 h.

**Figure 2 materials-15-08717-f002:**
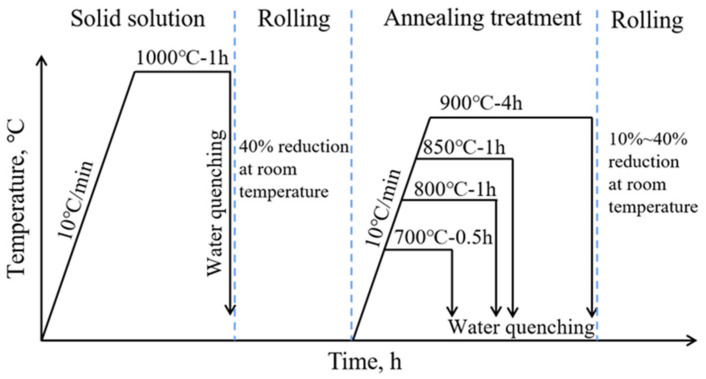
The schematic diagram of experimental processes.

**Figure 3 materials-15-08717-f003:**
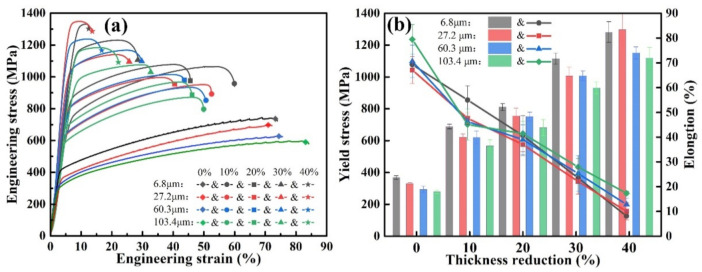
Tensile properties of Fe-18Mn-1.3Al-0.6C samples under different rolling reduction: (**a**) Engineering stress-strain curve; (**b**) Ultimate strength and elongation corresponding to (**a**).

**Figure 4 materials-15-08717-f004:**
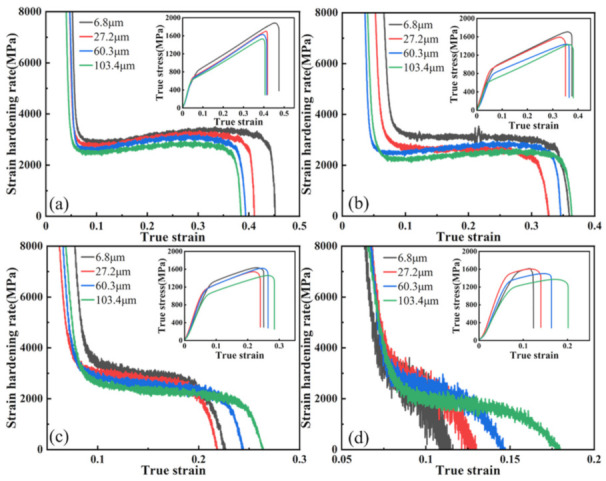
Strain hardening rate curves and true stress-strain curves of the studied samples at different reduction: (**a**) 10%; (**b**) 20%; (**c**) 30%; (**d**) 40%.

**Figure 5 materials-15-08717-f005:**
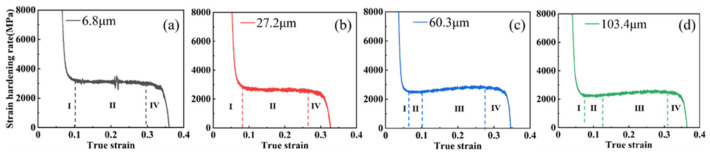
Work-hardening rate curves for each grain size samples at 20% reduction: (**a**) 6.8 μm; (**b**) 27.2 μm; (**c**) 60.3 μm; (**d**)103.4 μm.

**Figure 6 materials-15-08717-f006:**
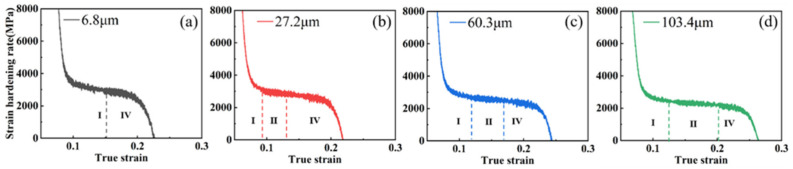
Work-hardening rate curves for each grain size sample at 30% reduction: (**a**) 6.8 μm; (**b**) 27.2 μm; (**c**) 60.3 μm; (**d**) 103.4 μm.

**Figure 7 materials-15-08717-f007:**
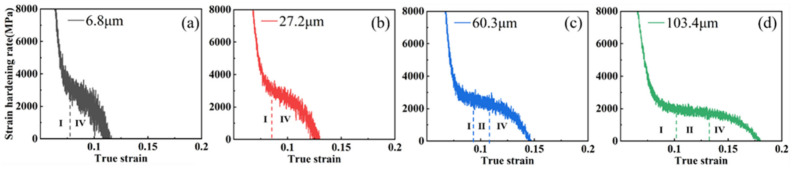
Work-hardening rate curves for each grain size sample at 40% reduction: (**a**) 6.8 μm; (**b**) 27.2 μm; (**c**) 60.3 μm; (**d**) 103.4 μm.

**Figure 8 materials-15-08717-f008:**
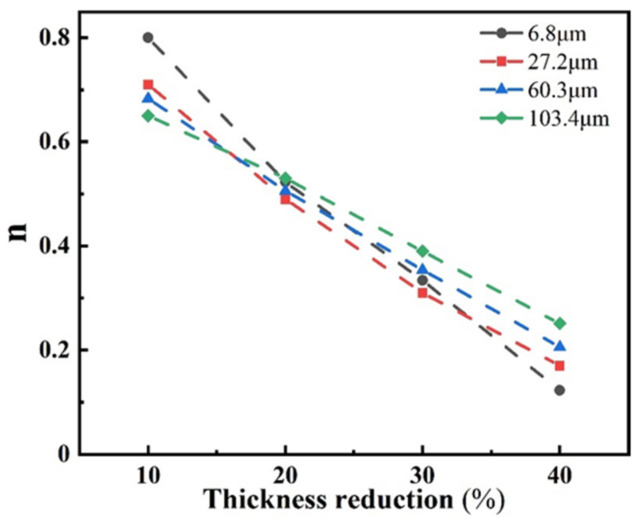
Relationship between work-hardening index *n* and the reduction of samples with different grain sizes.

**Figure 9 materials-15-08717-f009:**
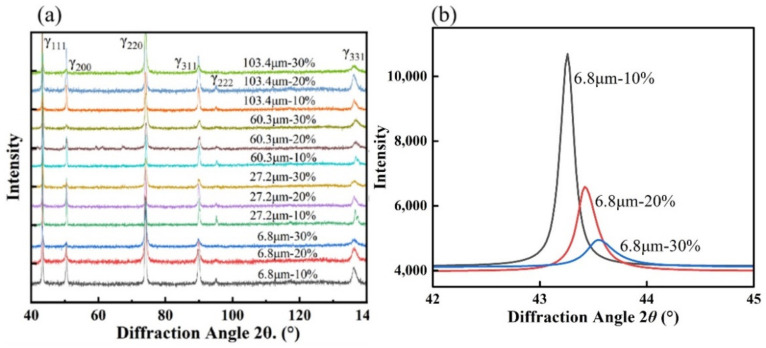
(**a**)XRD patterns of rolled specimens samples with different grain sizes at different reduction; (**b**) The (111) diffraction peaks of samples with the average grain size of 6.8 μm.

**Figure 10 materials-15-08717-f010:**
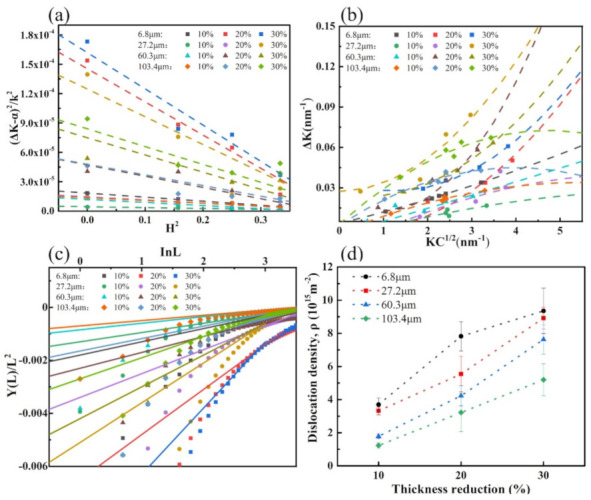
The relationship patterns of the experimental steels with different grain size under different reduction on mWH/WA method: (**a**) ${\left(\Delta K-\alpha \right)}^{2}/K^{2}$ and $H^{2}$; (**b**) $KC^{1/2}$ and ΔK; (**c**) $Y\left(L\right)/L^{2}$ and lnL. And (**d**) is the calculated dislocation density of each sample.

**Figure 11 materials-15-08717-f011:**
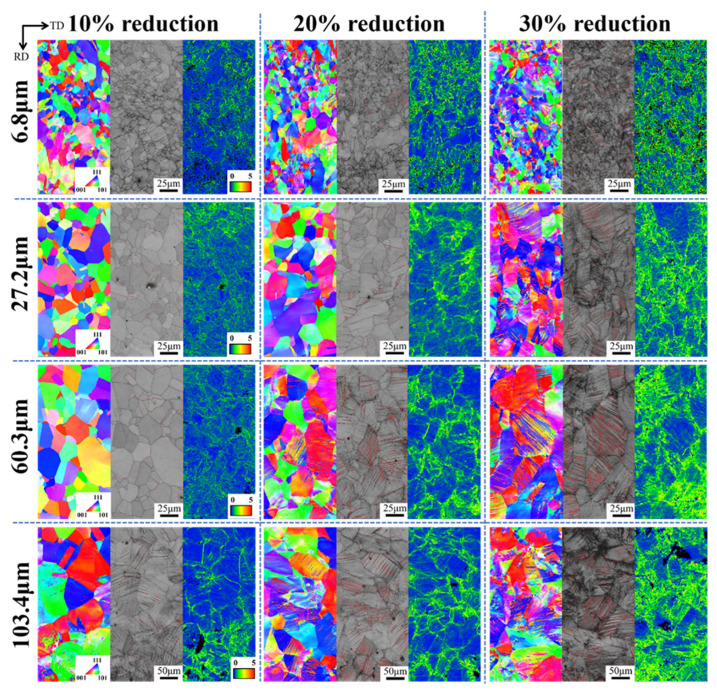
IPF, IQ (twin boundaries are marked by red lines) and KAM maps of annealed samples with different grain sizes after rolling.

**Figure 12 materials-15-08717-f012:**
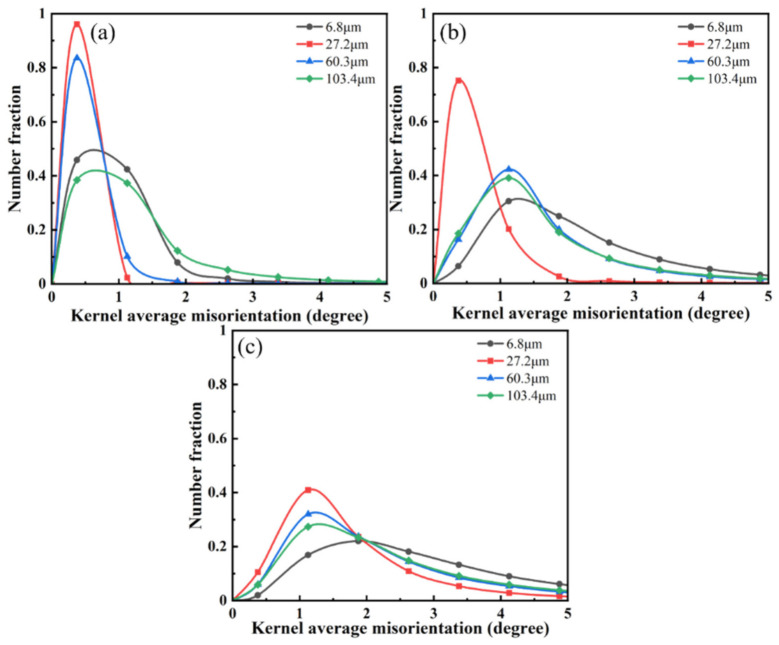
KAM distributions of annealed samples with different grain sizes under different reduction: (**a**) 10%; (**b**) 20%; (**c**) 30%.

## Data Availability

Not applicable.

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
