# Peer review of "Effect of Grain Size on the Plastic Deformation Behaviors of a Fe-18Mn-1.3Al-0.6C Austenitic Steel"

_materials, 2022, doi:10.3390/ma15248717_

Round 1

Reviewer 1 Report

The article presents the results of research on the effect of grain size on austenitic steel Fe-18Mn-1.3Al-0.6C on its susceptibility to plastic deformation. The presented research results are of a utilitarian nature when shaping the properties of TWIP steel.

The article examines the effect of grain size on twin behavior, dislocation proliferation and KAM distribution. This allows a generalization of their influence on the mechanical properties and behavior during curing. But my doubts are raised by the title of the article, because it does not allow for the development of the plastic deformation mechanism. Therefore, I believe that the title of the article should be changed.

Line 60: "Some researchers illustrated ..." references should be supplemented. References must also be provided in equation (1).

The materials and research methodology have been described in detail, but the grain size has not been determined. This should be stated in the test methodology and not in the test results. In addition, the authors used the Scanning Electron Microscope (SEM) for the research using the EBSD observation technique.

Line 109: “… corroded with 7% nitric ……….” Is the word "corroded" correct? For me, it's about digestion.

Line 119-120 "" In addition, few annealing twins were observed, ... "

 It should be marked in the photo where it can be observed.

Fig. 11: The drawing is difficult to read. The photos of the structure should be larger.

Author Response

Responses to Reviewer 1

The article presents the results of research on the effect of grain size on austenitic steel Fe-18Mn-1.3Al-0.6C on its susceptibility to plastic deformation. The presented research results are of a utilitarian nature when shaping the properties of TWIP steel.

The article examines the effect of grain size on twin behavior, dislocation proliferation and KAM distribution. This allows a generalization of their influence on the mechanical properties and behavior during curing. But my doubts are raised by the title of the article, because it does not allow for the development of the plastic deformation mechanism. Therefore, I believe that the title of the article should be changed.

Response: Thank you very much for your kind comments. According to your suggestions, the title of this article has been revised to be “Effect of grain size on the plastic deformation behaviors of a Fe-18Mn-1.3Al-0.6C austenitic steel”.

Line 60: "Some researchers illustrated ..." references should be supplemented. References must also be provided in equation (1).

Response: Many thanks for the careful comments. The missed references have been added in the new submission.

The materials and research methodology have been described in detail, but the grain size has not been determined. This should be stated in the test methodology and not in the test results. In addition, the authors used the Scanning Electron Microscope (SEM) for the research using the EBSD observation technique.

Response: Thank you for the comments. In the revised paper, the results of grain size (Fig. 2) have been moved to section 2. Experimental procedures. In addition, the statements on the Scanning Electron Microscope (SEM) were also added.

Line 109: “… corroded with 7% nitric ……….” Is the word "corroded" correct? For me, it's about digestion.

Response: Thank you for the careful comments. We are sorry for the mistakes. The word "corroded" has been corrected to be “etched” in the revised manuscript.

Line 119-120 "" In addition, few annealing twins were observed, ... "

 It should be marked in the photo where it can be observed.

 Response: Many thanks for your comments. In the revised paper, selected annealing twins have been marked in the figure.

Fig. 11: The drawing is difficult to read. The photos of the structure should be larger.

Response: Thank you so much for the comments. Fig. 11 has been enlarged in the new submission.

Reviewer 2 Report

High manganese austenitic steels are generating a lot of interest with potential applications for structural parts in the automotive industry due to their high strength. The authors of the paper "Effect of grain size on the plastic deformation mechanisms of a Fe-18Mn-1.3Al-0.6C austenitic steel" have investigated the influence of the grain size and rolling reduction on the steel’s plasticity. The authors provide an interesting experimental results. However, some points of the manuscript are very questionable. The paper is needed to be seriously modified accordingly following comments:

1.             The most of cited papers are too old. The state of art should be enriched with literature items from the recent (e.g., works of Pozdniakov et al devoted to high-Mn steels).

2.             The elastic range on the stress-strain curve seems to be too large. It is not possible to achieve in the crystalline metallic materials elastic elongation of 5 %. The authors did not use the extensometer during the tensile test. Such fact makes the obtained results (especially, elongation) very questionable.

3.             It is recommended to consider the influence of the grain size and thickness reduction on the yield strength. This parameter more significant of the practical usage than ultimate strength.

4.             The addition of the mechanical properties and microstructural analysis of undeformed sample significantly improves the scientific part of the paper by providing of the pure influence of the grain size on the deformation behavior during tension. It is recommended to add to the manuscript such information.

5.             It is not possible to achieve the dislocation density higher than 1016 cm-2. At such density the structure becomes fully amorphous. It is not possible to achieve amorphization by cold rolling to 20-30 %. This means that applied by the authors methods for determination of the dislocation density using XRD data are incorrect.

6.             Minor corrections:

-          Table 1 does not provide any information and may be removed.

-          How many samples were tested for each state? The information should be added to the manuscript.

-          Standard deviations should be added to the values of the microstructural parameters and mechanical properties.

-          The quality of the microstructural images should be improved.

Author Response

Responses to Reviewer 2

High manganese austenitic steels are generating a lot of interest with potential applications for structural parts in the automotive industry due to their high strength. The authors of the paper "Effect of grain size on the plastic deformation mechanisms of a Fe-18Mn-1.3Al-0.6C austenitic steel" have investigated the influence of the grain size and rolling reduction on the steel’s plasticity. The authors provide an interesting experimental results. However, some points of the manuscript are very questionable. The paper is needed to be seriously modified accordingly following comments:

Response: The English was polished by professional serves (i.e., MDPI English editing) according to your suggestions. The English-Editing-Certificate was uploaded.

  1. The most of cited papers are too old. The state of art should be enriched with literature items from the recent (e.g., works of Pozdniakov et al devoted to high-Mn steels).

Response: Thank you very much for your kind comments. Two references from Pozdniakov et al. have been citied. In addition, some old references have been replaced by new ones and few newly published papers were cited in the revised manuscript.

  1. The elastic range on the stress-strain curve seems to be too large. It is not possible to achieve in the crystalline metallic materials elastic elongation of 5 %. The authors did not use the extensometer during the tensile test. Such fact makes the obtained results (especially, elongation) very questionable.

Response: Thank you for the careful comments. In this study, the tensile tested samples were small, hence the extensometer in our lab can not be used during testing. It is true that the results enlarged the elongation. In the revised paper, the given elongation (Fig. 3 (b)) was corrected by considering the additional elastic elongation.

  1. It is recommended to consider the influence of the grain size and thickness reduction on the yield strength. This parameter more significant of the practical usage than ultimate strength.

Response: Thank you for the comments. According to your suggestion, the ultimate strength was replaced by yield strength in Fig. 3 (b) of the revised paper.

  1. The addition of the mechanical properties and microstructural analysis of undeformed sample significantly improves the scientific part of the paper by providing of the pure influence of the grain size on the deformation behavior during tension. It is recommended to add to the manuscript such information.

Response: Many thanks for your comments. Your suggestion is nice. We added the results of undeformed sample (0%) in the revised paper.

  1. It is not possible to achieve the dislocation density higher than 1016cm-2. At such density the structure becomes fully amorphous. It is not possible to achieve amorphization by cold rolling to 20-30 %. This means that applied by the authors methods for determination of the dislocation density using XRD data are incorrect.

Response: Thank you so much for the comments. We checked the original data and calculation process and found some mistake to over estimate the dislocation density. The corrected data were given in the revised paper. For TWIP steels, the dislocation density was easily increased to about 1015 cm-2 from previous studies.

  1. Minor corrections:

-          Table 1 does not provide any information and may be removed.

-          How many samples were tested for each state? The information should be added to the manuscript.

-          Standard deviations should be added to the values of the microstructural parameters and mechanical properties.

-          The quality of the microstructural images should be improved.

Response: Thank you very much for your kind comments. Table 1 was removed in the revised manuscript. Three samples were tested for each state and the additional illustration was given. Standard deviations of the microstructural parameters and mechanical properties were given in the new submission. The quality of the microstructural images was also improved in the new paper.

Reviewer 3 Report

Dear authors. Thank you very much for the article. My overall opinion is that article quality is very high and can help other researchers in the research work. However i Will be very grateful if you will add Some informations which maily are:

1. Can u please give results for statistically significant differences in variance and mean? With ANOVA tool or with Excel functions ftest (test for variance) and ttest (test for mean)

2. Can you please describe testing Machine details such as Manufacturer, Limits of Machine, Strain measurement technique, accuracy of Strain measurement, accuracy of stress measurement)

3. Other not main question if u measured also Some type of hardness (Vickers, Brinell, Rockwell or microhardness) or if u measured also Some type of toughness or fatigue test

4. Please give informations about impact of better studied mechanical properties on the applicability of such material (in which areas the improvement of tested properties leads to better material applicability).

Once again thank you for the article. Wish you good luck. 

Author Response

Responses to Reviewer 3

Dear authors. Thank you very much for the article. My overall opinion is that article quality is very high and can help other researchers in the research work. However i Will be very grateful if you will add Some informations which maily are:

  1. Can u please give results for statistically significant differences in variance and mean? With ANOVA tool or with Excel functions ftest (test for variance) and ttest (test for mean)

Response: Thank you very much for your kind comments. It is true different statistical approaches could result in small variations of the final results. In the revised paper, Three samples were tested for each state and the average results and standard deviations of the microstructural parameters and mechanical properties were given to indicate and minimize such differences.

  1. Can you please describe testing Machine details such as Manufacturer, Limits of Machine, Strain measurement technique, accuracy of Strain measurement, accuracy of stress measurement)

Response: Many thanks for your comments. According to you good suggestion, the information on the testing machine were added in the new submission.

  1. Other not main question if u measured also Some type of hardness (Vickers, Brinell, Rockwell or microhardness) or if u measured also Some type of toughness or fatigue test

Response: Thank you for the careful comments. In this work, only tensile property was checked to mainly investigate the work hardening behaviors. We did not measure the hardness, impact toughness and fatigue.

  1. Please give informations about impact of better studied mechanical properties on the applicability of such material (in which areas the improvement of tested properties leads to better material applicability). 

Once again thank you for the article. Wish you good luck. 

Response: Thank you so much for the comments. According to you suggestion, we added the discussions on the impact of better studied mechanical properties on the applicability of such material in the revised manuscript.

Round 2

Reviewer 2 Report

The authors have answered previous comments and added necessary  information to the manuscript. The paper may be accepted for publication.